# Disulfides from the Brown Alga *Dictyopteris Membranacea* Suppress M1 Macrophage Activation by Inducing AKT and Suppressing MAPK/ERK Signaling Pathways

**DOI:** 10.3390/md18110527

**Published:** 2020-10-24

**Authors:** Maria G. Daskalaki, Paraskevi Bafiti, Stefanos Kikionis, Maria Laskou, Vassilios Roussis, Efstathia Ioannou, Sotirios C. Kampranis, Christos Tsatsanis

**Affiliations:** 1Laboratory of Clinical Chemistry, School of Medicine, University of Crete, 70013 Heraklion, Greece; m.daskalaki@med.uoc.gr (M.G.D.); vivi.mpafiti@gmail.com (P.B.); maria.laskou95@gmail.com (M.L.); 2Institute of Molecular Biology and Biotechnology, FORTH, 71100 Heraklion, Greece; 3Section of Pharmacognosy and Chemistry of Natural Products, Department of Pharmacy, National and Kapodistrian University of Athens, Panepistimiopolis Zographou, 15771 Athens, Greece; skikionis@pharm.uoa.gr (S.K.); roussis@pharm.uoa.gr (V.R.); eioannou@pharm.uoa.gr (E.I.); 4Section of Plant Biochemistry, Department of Plant and Environmental Sciences, University of Copenhagen, Thorvaldsensvej 40, 1871 Frederiksberg, Denmark

**Keywords:** *Dictyopteris membranacea*, disulfides, macrophages, inflammation, TNFα, iNOS, IL-6, AKT, ERK1/2

## Abstract

Inflammation is part of the organism’s response to deleterious stimuli, such as pathogens, damaged cells, or irritants. Macrophages orchestrate the inflammatory response obtaining different activation phenotypes broadly defined as M1 (pro-inflammatory) or M2 (homeostatic) phenotypes, which contribute to pathogen elimination or disease pathogenesis. The type and magnitude of the response of macrophages are shaped by endogenous and exogenous factors and can be affected by nutrients or therapeutic agents. Multiple studies have shown that natural products possess immunomodulatory properties and that marine algae contain products with such action. We have previously shown that disulfides isolated from *Dictyopteris membranacea* suppress nitric oxide (NO) production from activated macrophages, suggesting potential anti-inflammatory actions. In this study, we investigated the anti-inflammatory mechanism of action of bis(5-methylthio-3-oxo-undecyl) disulfide (**1**), 5-methylthio-1-(3-oxo-undecyl) disulfanylundecan-3-one (**2**) and 3-hexyl-4,5-dithiocycloheptanone (**3**). Our results showed that all three compounds inhibited M1 activation of macrophages by down regulating the production of pro-inflammatory cytokines TNFα, IL-6 and IL-12, suppressed the expression of the NO converting enzyme iNOS, and enhanced expression of the M2 activation markers Arginase1 and MRC1. Moreover, disulfides **1** and **2** suppressed the expression of glucose transporters GLUT1 and GLUT3, suggesting that compounds **1** and **2** may affect cell metabolism. We showed that this was due to AKT/MAPK/ERK signaling pathway modulation and specifically by elevated AKT phosphorylation and MAPK/ERK signal transduction reduction. Hence, disulfides **1**–**3** can be considered as potent candidates for the development of novel anti-inflammatory molecules with homeostatic properties.

## 1. Introduction

Macrophages are professional phagocytes, deriving from bone marrow monocyte precursors, and can be found in all tissues as circulating or tissue resident macrophages. They function as antigen-presenting cells (APCs) after they recognize and internalize a wide variety of phagocytic targets, such as pathogens, apoptotic or necrotic cells, mirroring the diverse macrophage function [1,2]. Macrophages, along with mast cells, neutrophils, and NK cells, are members of the innate immune defense mechanism, participating both in the initiation and resolution of inflammation, responding to inflammatory signals and triggering T helper cell responses. Depending on the environmental stimuli and stage of the immune response, macrophages obtain different activation phenotypes broadly defined as classically activated (M1) and alternatively activated (M2) macrophages, both consisting of an array of phenotypes [3]. M1-activated macrophages respond to lipopolysaccharides (LPS) and/or IFN-γ and express pro-inflammatory cytokines, such as TNFα, IL-6, IL-12, and nitric oxide (NO), while abrogating Th1 pro-inflammatory responses and contributing to pathogen clearance. Alternatively activated macrophages (M2) are induced by anti-inflammatory cytokines IL-4 and IL-10, TGF-β production, immunocomplexes and glucocorticoids and express resolution factors, such as IL-10, Arginase 1, Fizz1, MRC1, and Ym1, mediating Th2 anti-inflammatory responses and contributing to parasitic infection clearance, tissue remodeling, angiogenesis, and wound healing [3,4]. In response to the various stimuli mentioned above, many signaling pathways are activated, resulting in transcriptional and epigenetic regulation of target genes. Among those, the PI3K/AKT signaling cascade plays a central role in macrophage polarization, cell survival, proliferation, as well as cellular metabolism [4,5,6,7]. PI3K/AKT signaling is activated downstream of multiple cell-surface receptors, such as pathogen recognition receptors like TLR4, cytokines, chemokines and Fc receptors [4], regulating downstream targets and cross-talking with other important signal transduction networks, such as Mitogen-activated protein kinase (MAPK)/extracellular signal-regulated kinase 1/2 (ERK1/2) signaling [8]. Fine-tuned crosstalk between PI3K/AKT and MAPK/ERK signaling is essential for cell homeostasis and is achieved through a feedback control loop, where activated ERK inhibits AKT phosphorylation and vice versa [9].

In recent years, intensive efforts in drug discovery research has led to the increased interest in the isolation of health-promoting compounds from marine organisms, such as seaweeds and microalgae. More than 1500 brown algae (Ochrophyta) are distributed worldwide, producing a wide pool of bioactive metabolites with anti-inflammatory, anticancer, antioxidant, antimicrobial and other properties [10]. *Dictyopteris membranacea* (synonym *Dictyopteris polypodioides*) is a species of brown macroalgae, growing in shallow waters along the Atlantic coast, Asia, and the Mediterranean sea, known for its odoriferous character mainly due to derivatives of C_11_-hydrocarbons [11].

We have previously shown that disulfides isolated from *D. membranacea* exhibit strong anti-inflammatory properties in LPS activated macrophages [12]. In view of the increasing need for new pharmaceutical agents to fight chronic inflammatory pathologies and acute inflammatory deregulations, we aimed to investigate further the anti-inflammatory properties of these disulfides. In this study, we illustrate the in vitro mechanism of the function of disulfides **1**–**3** (Figure 1A) and conclude that all three impair classical M1 macrophage activation by interfering in AKT/MAPK/ERK signal transduction cross-talk.

## 2. Results and Discussion

### 2.1. Isolation of Metabolites **1**–**3** and Evaluation of Their Anti-Inflammatory Potency and Toxicity

We have previously reported the isolation of bis(5-methylthio-3-oxo-undecyl) disulfide (**1**), 5-methylthio-1-(3-oxo-undecyl)disulfanylundecan-3-one (**2**) and 3-hexyl-4,5-dithiocycloheptanone (**3**) from a population of *D. membranacea* collected in Gerolimenas Bay, Greece, and that compounds **2** and **3** exhibit significant anti-inflammatory activity by suppression of NO release from macrophages, while the anti-inflammatory activity of **1** was not evaluated since it was isolated in minute amounts [12]. Chemical investigation of another population of *D. membranacea* collected from Tinos island, Greece, allowed for the re-isolation of **1**–**3** in sufficient amounts for thorough evaluation of their anti-inflammatory action.

In order to determine the anti-inflammatory potency of **1**–**3**, we treated RAW 264.7 murine macrophages with increasing concentrations of the metabolites in macrophages activated for 48 h using LPS. Subsequently, we measured the secreted NO using the Griess reaction and compared it to carbowax 400-treated cells (the solvent used to dissolve the metabolites). The results showed that **1** exhibited IC_50_ = 2.6 ± 0.35 μΜ, while **2** and **3** displayed IC_50_ = 3.8 ± 0.29 μΜ and IC_50_ = 14.2 ± 1.67 μΜ, respectively (Figure 1B). In order to determine whether the anti-inflammatory outcome is the result of the cytostatic potential of the compounds, we conducted a cell viability and proliferation test treating RAW 264.7 macrophages with the tetrazolium dye MTT. We confirmed our earlier findings for compounds **2** and **3** and showed that the cytostatic potential of compound **1** was above 62.5 μΜ at 48 h following treatment and above 7.8 μΜ at 72 h (Figure 1C–E), suggesting that lower concentrations of compound **1** could be optimal for pharmaceutical use. The findings are summarized in Table 1.

### 2.2. Macrophages Exposed to Compounds **1**–**3** Suppress Expression of Pro-Inflammatory Genes and Alter Metabolic Gene Expression

Macrophages become activated in response to various signals obtaining different activation phenotypes expressing different levels of pro- and anti-inflammatory cytokines, signaling and cell surface molecules, and are characterized as M1 or M2 type [13]. The RAW 264.7 murine cell line is commonly used to test anti-inflammatory properties of natural products, as it is a reliable in vitro macrophage-like model. RAW 264.7 cells express inflammatory mediators in response to stimuli, simulating inflammatory responses. To identify the mechanism of action of disulfides **1**–**3**, we treated RAW 264.7 macrophages with the compounds for 12, 24 or 48 h in a concentration approximately three-times higher than the IC_50_ value of each compound and then measured the expression pattern of inflammatory mediators (Figure 2).

First, we measured the effect of compounds **1**–**3** on the expression of iNOS (inducible nitric oxide synthase), the enzyme that catalyzes production of NO, and the pro-inflammatory cytokine TNFα, both markers of M1 activation, in non-activated macrophages. We showed that all three compounds suppressed production of TNFα in naïve macrophages 24 h following exposure to the compounds (Figure 2A). Furthermore, iNOS expression was reduced up to 35% in cells exposed to the compounds, both 24 and 48 h following exposure (Figure 2B). M2 polarization is characterized by the expression of molecules, such as the enzymes Arginase 1, MRC1, Fizz1, the transcription factor c/EBPβ, and anti-inflammatory cytokines, such as IL-10 [4]. In this context, we tested the expression profile of major M2 markers that are signatures of anti-inflammatory phenotype in RAW 264.7 macrophages exposed to the compounds. We showed that only disulfides **1** and **2**, which also exhibit structural similarities, up-regulate the expression of the M2 marker Arginase 1 (Figure 2C), following 48 h exposure. In addition, disulfides **1** and **2** up-regulated the expression of MRC1, both at 24 and 48 h, whereas disulfide **3** moderately up-regulated MRC1 expression 24 h following exposure (Figure 2D). All three compounds showed no significant effect on the expression of IRAK-M (Figure 2E), a monocyte-exclusive non-kinase protein that negatively regulates Toll-like receptor signaling and contributes to an anti-inflammatory phenotype [14,15].

Glycolysis rapidly converts glucose to pyruvate and generates two molecules of ATP. Macrophage activation is a highly energetic process. Phagocytosis, reactive oxygen species production (ROS), and pro-inflammatory cytokine production are highly dependent on glycolysis [16]. Glucose uptake by different cell types is facilitated by glucose transporters (GLUT). Macrophages express three glucose transporter genes, GLUT-1, GLUT-3 and GLUT-5. Among them, GLUT-1 is the most abundant and rate limiting, and its overexpression in RAW 264.7 macrophages leads to hyperinflammation [17]. GLUT 1 and GLUT 3 expression are induced by different signals such as IGF1 or cytokines, leading to NFkB activation [18]. M1 activation relies heavily on glycolysis; however, the transition to the M2 phenotype requires a shift to fatty acid oxidation (FAO) [19]. IL-4 treatment polarizes macrophages to an M2-like phenotype and induces FAO [20].

Subsequently, we tested the expression profile of key metabolic genes that have been shown to contribute to different macrophage activation phenotypes. We treated naïve RAW 264.7 macrophages with each of the three compounds for 12, 24 and 48 h and measured mRNA levels of selected metabolic genes. We showed that both disulfides **1** and **2** down-regulated the expression of GLUT-1 (Figure 3A), the main glucose transporter of macrophages which facilitates glucose uptake, essential for the pro-inflammatory M1 phenotype, potentially reducing glucose availability and glycolytic capacity of the cells. Moreover, disulfide **1**, the one with the lowest IC_50_ value, was found to also reduce GLUT-3 mRNA expression, the second most abundant glucose transporter in macrophages, 48 h following exposure to the compound (Figure 3B). Importantly, we found that the three compounds did not exhibit any significant difference in the expression pattern of the PPAR gamma Co-activator-1 alpha (PGC1α) (Figure 3C), a mediator of FAO, indicating that any M2 polarization functions of the compounds are likely FAO-independent.

### 2.3. Compounds **1**–**3** Impair TLR4-Mediated Macrophage Activation

In order to determine the effect of disulfides **1**–**3** on the activation potential of macrophages, we pre-treated RAW 264.7 macrophages with each of the three compounds for 1 h and then stimulated cells with *Escherichia coli*-derived LPS for 12, 24 or 48 h. This type of stimulation triggers TLR4-mediated activation and induction of M1 mediators, whereas prolonged incubation with LPS induces an M2-like phenotype termed as endotoxin tolerance, characterized by expression of anti-inflammatory markers. We measured mRNA levels of the pro-inflammatory marker iNOS (Figure 4A) and showed that all three compounds significantly down-regulated its expression both 12 and 24 h post stimulation. Similarly, we showed that pro-inflammatory cytokines IL-6 and IL-12b were significantly lower in all three compound-treated groups (Figure 4B,C) compared to cells treated with LPS alone. To further elucidate the anti-inflammatory properties of disulfides **1**–**3** in activated macrophages, we measured protein levels of secreted pro-inflammatory cytokines using ELISA in compound-treated macrophages that had been stimulated with LPS for 12 or 24 h. All three compound-treated macrophages secreted significantly less TNFα 24 h following stimulation, whereas macrophages treated with compounds **1** and **2** secreted less TNFα even 12 h following stimulation (Figure 4D). In addition, all compound-treated macrophages secreted significantly less IL-6 24 h following LPS stimulation (Figure 4E). It has been previously shown that disulfides, such as defensins, can directly interact with LPS acting as scavengers, neutralizing its action [21]. Nevertheless, it is less likely that it is the case for disulfides **1**–**3**, since their anti-inflammatory properties are also evident in non-stimulated cells (Figure 2).

Subsequently, we aimed to identify the effect of compounds **1**–**3** on the expression of M2-like anti-inflammatory markers 24 and 48 h following LPS stimulation. We showed that although M1 activation was impaired in compound-treated cells, M2 markers were induced upon LPS stimulation at 24 and 48 h as a negative feedback mechanism. All three disulfides did not affect the expression of the M2 marker Arginase 1 (Figure 4F), whereas MRC1 was significantly up-regulated in activated macrophages treated with disulfides **1** and **2** (Figure 4G). IRAK-M expression is induced in macrophages following LPS activation to suppress TLR4 signals and promote M2-like responses. In cells treated with all three compounds, IRAK-M expression was significantly reduced, suggesting that suppression of M1 responses did not allow induction of IRAK-M and that IRAK-M is not a direct target of the anti-inflammatory action of these compounds (Figure 4F).

The anti-inflammatory action of the compounds was confirmed in primary macrophages. For this purpose, we pretreated thioglycolate-elicited mouse macrophages for 1 h with the respective compound and then activated them with LPS for 12 h. The mRNA expression levels of the pro-inflammatory markers iNOS, IL-6 and IL-12 was measured (Figure 5B–D) and showed that all three compounds downregulated their expression, confirming the compounds’ anti-inflammatory properties. In addition, NO secretion was significantly reduced in the cell culture medium, further confirming the anti-inflammatory action of the compounds.

### 2.4. The Effect of Compounds **1**–**3** in AKT and MAPK/ERK Signaling

MAPK signaling pathway regulates a number of cellular processes, such as cell proliferation, survival, apoptosis, and transcription, indicating that MAPK signaling cascade is essential for a number of cellular processes, including immunity and metabolism [22]. Deregulation of ERK 1/2 signaling has been implicated in several inflammatory pathologies, such as rheumatoid arthritis, psoriatic arthritis and inflammatory bowel diseases, e.g., Crohn’s disease [23]. In addition, the PI3K/AKT/mTOR signaling pathway is activated upon several receptor signals, such as insulin and cytokines mediating inflammatory and metabolic processes orchestrating the inflammatory response [4]. AKT signaling is known to down-regulate inflammatory responses, promoting M2-like responses and inhibiting M1 activation, negatively regulating TLR and NF-κΒ signaling [7,24,25].

Consequently, in order to further elucidate the molecular mechanism underlying the anti-inflammatory activity of disulfides **1**–**3**, we pre-treated RAW 264.7 macrophages with each compound and then activated them with 100 ng/mL LPS for 20, 30 and 60 min. Total lysates were analyzed by Western blot and the phosphorylation levels of ERK 1/2 and AKT were detected. It was observed that all disulfides **1**–**3** negatively regulated basal phosphorylation levels of ERK 1/2 (Figure 6B,E), coming into agreement with the reduced levels of pro-inflammatory mediators in compound-treated naïve macrophages. Notably, no significant differences were observed in LPS-activated macrophages. In addition, basal phosphorylation levels of AKT (at serine 473) were found to be elevated in all three compound-treated cells, indicating that cells are prompted to an M2-like phenotype (Figure 6C,F). Disulfides **2** and **3** augmented LPS-induced phosphorylation of AKT (Figure 5F), suggesting that they promoted anti-inflammatory responses. At the same time disulfides **2** and **3** suppressed pro-inflammatory signals, such as ERK1/2, which is triggered by LPS.

## 3. Materials and Methods

### 3.1. General Experimental Procedures

Optical rotations were measured on a Krüss polarimeter (A. KRÜSS Optronic GmbH, Hamburg, Germany) equipped with a 0.5 dm cell. UV spectra were recorded on a Lambda 40 UV/Vis spectrophotometer (Perkin Elmer Ltd., Beaconsfield, UK). IR spectra were obtained on an Alpha II FTIR spectrometer (Bruker Optik GmbH, Ettlingen, Germany). Low-resolution EI mass spectra were measured on a Thermo Electron Corporation DSQ mass spectrometer (Thermo Electron Corporation, Austin, TX, USA) using a Direct-Exposure Probe (Thermo Electron Corporation, Austin, TX, USA). NMR spectra were recorded on a DRX 400 spectrometer (Bruker BioSpin GmbH, Rheinstetten, Germany). The 2D experiments (HSQC, HMBC, COSY, NOESY) were performed using standard Bruker pulse sequences. Column chromatography separations were performed with Kieselgel 60 (Merck, Darmstadt, Germany). HPLC separations were conducted on a Pharmacia LKB 2248 liquid chromatography pump (Pharmacia LKB Biotechnology, Uppsala, Sweden) equipped with a RI-102 Shodex refractive index detector (ECOM spol. s r.o., Prague, Czech Republic) using an Econosphere Silica 10 μm (250 × 10 mm i.d.; Grace, Columbia, MD, USA) column. TLC were performed with Kieselgel 60 F_254_ aluminum plates (Merck, Darmstadt, Germany) and spots were detected after spraying with 20% H_2_SO_4_ in MeOH reagent and heating at 100 °C for 1 min.

### 3.2. Collection of Algal Material

Specimens of *D. membranacea* were collected by hand at Agios Sostis on Tinos island, Greece, at a depth of 0.5 to 2 m in June 2017. A voucher specimen of the alga has been deposited at the Herbarium of the Section of Pharmacognosy and Chemistry of Natural Products, Department of Pharmacy, National and Kapodistrian University of Athens (ATPH/MP0583).

### 3.3. Isolation of Metabolites **1**–**3**

Specimens of the fresh alga were exhaustively extracted with mixtures of CH_2_Cl_2_/MeOH at room temperature. Evaporation of the solvents in vacuo afforded a dark green oily residue (8.5 g) that was subjected to vacuum column chromatography on silica gel, using cyclohexane with increasing amounts of EtOAc, followed by EtOAc with increasing amounts of MeOH as the mobile phase, to yield 11 fractions (A–K). Fraction B (20% EtOAc in cyclohexane, 0.85 g) was fractionated by gravity column chromatography on silica gel, using cyclohexane with increasing amounts of EtOAc as the mobile phase, to yield 9 fractions (B1–B9). Fraction B4 (5% EtOAc in cyclohexane, 135.3 mg) was purified by normal-phase HPLC, using cyclohexane/EtOAc (96:4) as eluent, to yield **3** (60.1 mg). Fraction B5 (7% EtOAc in cyclohexane, 77.2 mg) was purified by normal-phase HPLC, using cyclohexane/EtOAc (95:5 and subsequently 93:7) as eluent, to yield **1** (7.5 mg), **2** (4.1 mg) and **3** (3.5 mg).

### 3.4. Cell Culture and Compound Dilutions

Mouse macrophage cell line RAW 264.7 was cultured using DMEM medium (Gibco, 21885-025) supplemented with 10% heat inactivated Fetal Bovine Serum (Gibco, 10270-106) and 1% penicillin streptomycin (Gibco, 15070-063) at 37 °C in the presence of 5% CO_2_. Each compound used was diluted in CarbowaxTM 400 + 10% abs. ethanol (E/0650DF/17, Fisher chemical), serving also as the control solvent. The final concentration of ethanol in the culture was 0.01% and of CarbowaxTM 400 was 0.1% indispensably of the compound dilution used. Macrophage activation was performed using 100 ng/mL LPS (L2630, Sigma) and in the case of compound-treated cells, macrophages were pre-treated for 1 h with the respective compound before LPS stimulation. The compounds’ concentrations used for macrophage treatments in Figure 2, Figure 3, Figure 4 and Figure 5 were 15.62 μΜ for compounds **1** and **2** and 31.25 μΜ for compound **3.**

### 3.5. Nitric Oxide Measurement

Here, 30 × 10^4^ RAW 264.7 mouse macrophages per sample were cultured overnight in 24-well plates and then pre-treated for 1 h with the respective concentrations for each compound. Then, macrophages were activated using 100 ng/mL LPS (L2630, Sigma) for 48 h. The amount of nitrite, an oxidative product of NO, released in each culture supernatant was measured using Griess reaction. Next, 50 μL of sulfanilamide solution (1% sulfanilamide in 5% H_3_PO_4_) was added to 50 μL of cell culture supernatant and the mix was incubated for 5 min at room temperature. Then, 50 μL of NED solution (0.1% N-1-napthylethylenediamine dihydrochlorite in H_2_O) was added and the absorbance was measured at 540 nm using an automated microplate reader (Infinite 200 PRO, Tecan). All incubations were performed in the dark and the nitrite concentration was appraised using a sodium nitrite standard curve.

### 3.6. MTT Cell Metabolism Measurement

Here, 5 × 10^3^ RAW 264.7 mouse macrophages were seeded in a 96-well plate (one plate per measurement) and cultured overnight. Then, the cells were treated with the respective compound concentration and incubated for 24, 48 or 72 h. The number of cells was measured prior to treatment and used as normalization control. Thiazolyl Blue Tetrazolium Bromide (MTT) (A2231.001, Applichem) was added to the cells in a final concentration of 500 μg/mL and then cells were incubated at 37 °C plus 5% CO_2_ for 4 h. The supernatant was removed, and cells were lysed in a mix of isopropanol with 0.4% HCl. Absorbance of each sample was measured using an automated microplate reader (Infinite 200 PRO, Tecan) at 600 nm. The average OD of each treated sample was normalized to the OD of the control sample and statistical analysis was performed using Graphpad Prism 7.0.

### 3.7. RNA Extraction, cDNA Synthesis and Quantitative PCR

Here, 25 × 10^4^ RAW 264.7 macrophages were seeded in a 24-well plate and cultured overnight in 0.5 mL DMEM. Cells were treated with the appropriate compound and activated using LPS for 12, 24 or 48 h. Wells were washed with ice-cold PBS and cells were harvested using Trizol reagent (Invitrogen, 15596026). RNA was either isolated immediately according to the manufacturer’s instructions or samples were stored in −80 °C. Total RNA (1 µg per sample) was reversed transcribed using PrimeScript™ 1st strand cDNA synthesis Kit (Takara, 6110A) and random 6-mer primers, according to the manufacturer’s instructions. Each sample was diluted five times and used as template in duplicates for two-step quantitative PCR in a 7500 Fast Real-Time PCR Instrument (Applied Biosystems^®^, 4351106) with 96-well Block Module as follows: start step 95 °C for 3 min and then 40 cycles of 95 °C for 10 s and 60 °C for 30 s. Amplification was performed using KAPA SyBr^®^ Fast Universal qPCR kit (Kapa Biosystems, KK4618). The primers used are listed in Appendix A. Data analysis was accomplished using mRNA levels expressed as relative quantification (RQ) values, which were calculated as RQ = 2(-DDCt), where DCt is (Ct (gene of interest)—Ct (housekeeping gene)). Rsp9 mRNA was used as the internal control gene.

### 3.8. ELISA

Cytokine concentration of mouse TNFα (Elisa Max™ Delux Set BioLegent, 430904) and mouse IL-6 (Elisa Max™ Delux Set BioLegent, 431301) in the medium of activated macrophages was determined following the manufacturer’s instructions.

### 3.9. Western Blot

Here, 40 × 10^4^ RAW 264.7 macrophages were seeded in a 12-well plate and pre-treated for 1 h with each compound, then cells were activated using 100 ng/mL LPS for 20, 30 or 60 min. Cells were washed with 1x PBS and cell protein lysates were collected in radio-Immune Precipitation Assay buffer (RIPA, 10 mM Tris-HCl (pH 8), 10 mM EDTA (pH 8), 140 mM NaCl, 1% Triton-X, 1% Na-deoxycolate, 0.1% SDS) containing phosphatase and protease inhibitors (Complete Protease Inhibitor Cocktail, Roche, Basel, Switzerland). Protein samples were sonicated 3 times for 20 sec and then protein concentration was quantified using Bicinchonic acid kit (BCA). Then, 15 μg of total protein was electrophoresed in 12% denaturing polyacrylamide gel and then wet transferred to 0.2 mm PVDF membrane (Macherey-Nagel, Germany) for 1 h at 400 mA. After blocking for 1 h with 5% BSA in PBS-T (pH 7.4) at room temperature, the membranes were incubated over night at 4 °C with primary antibody anti-*p*-AKT(Ser473) (#4060,Cell Signaling), anti-AKT (#4691,Cell Signaling), anti-*p*-p44/42 MAPK Erk1/2 (#4370, Cell Signaling), anti-p44/42 MAPK Erk1/2 (#9102,Cell Signaling), anti-tubulin (clone 1A2, Sigma-Aldrich). The anti-mouse and anti-rabbit secondary antibodies (Enzo Life Sciences) were incubated at room temperature for 1 h in 5% BSA. Visualization occurred using ECL system (Luminata, Classico, Merck Millipore) and a ChemiDoc XRS+ System (BioRad). Quantitation of protein was performed by band intensity using Image Lab Software (Bio-rad).

### 3.10. Isolation of Primary Peritoneal Mouse Macrophages

Here, 4% *w/v* thioglycollate Medium (Brewer) was diluted in normal saline, autoclaved, aliquoted and stored at 4 °C. Briefly, 8-week C57BL/6 mice were injected intraperitoneally with 1.5 mL thioglycollate medium and 3 days post injection mice were sacrificed. Macrophages were harvested by performing peritoneal lavage. Cells were washed with DMEM medium (Gibco, 21885-025) supplemented with 10% heat inactivated Fetal Bovine Serum (Gibco, 10270-106) and 1% penicillin streptomycin (Gibco, 15070-063) and seeded in 24-well plates in a cell density of 5 × 10^5^ cells. C57BL/6 mice were kept in pathogen-free animal facility in the Medical School of the University of Crete, Heraklion, Crete. All procedures were conducted in compliance with protocols approved by the Animal Care Committee of the University of Crete, School of Medicine (Heraklion, Crete, Greece) and the Veterinary Department of the Region of Crete (Heraklion, Crete, Greece), license number 269884/2018.

### 3.11. Statistical Analysis

All data are presented as mean ± SEM. Statistical analysis was performed using Graphpad Prism 7.0. A Mann–Whitney t-test was performed to test statistical analysis of each treated sample to control. Tukey’s test was performed to test statistical significance between groups, confirming the results. Differences with a *p* value < 0.05 are considered significant (* indicates *p* < 0.05, ** indicates *p* < 0.01, *** indicates *p* < 0.001).

## 4. Conclusions

Overall, our study showed that disulfides **1**–**3** possess anti-inflammatory properties by suppressing M1-type activation of macrophages and inducing genes that shift their activation to the M2-like phenotype. The compounds also suppressed the pro-inflammatory signaling cascade of MAPK, while inducing the anti-inflammatory signaling cascade of AKT, providing an additional potential mechanism for their anti-inflammatory action. Further studies are required to confirm these mechanisms. In conclusion, the results highlight the potential of disulfides **1**–**3** isolated from *D. membranacea* to be used as anti-inflammatory agents or as leads for the design of novel anti-inflammatory therapies.

## Figures and Tables

**Figure 1 marinedrugs-18-00527-f001:**
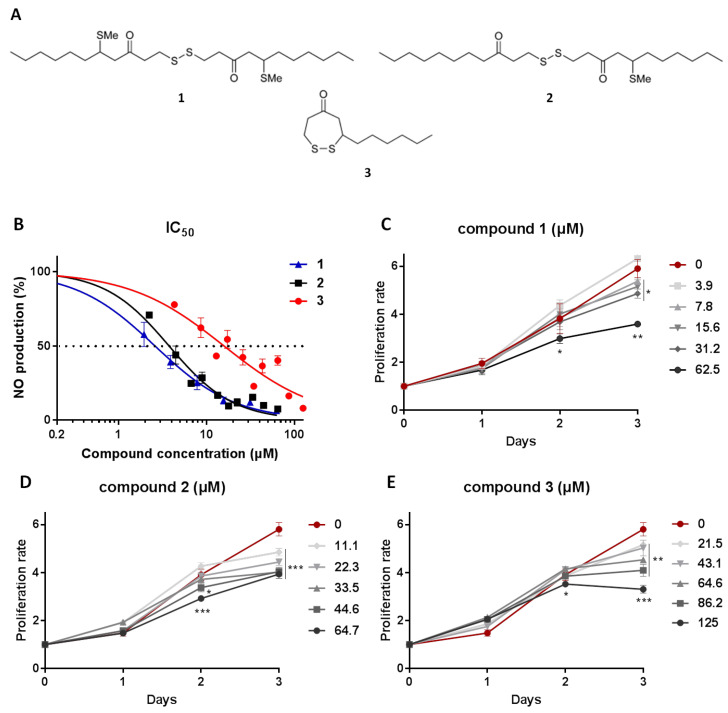
(**A**) Chemical structures of bis(5-methylthio-3-oxo-undecyl) disulfide (**1**), 5-methylthio-1-(3-oxo-undecyl)disulfanylundecan-3-one (**2**) and 3-hexyl-4,5-dithiocycloheptanone (**3**) isolated from *D. membranacea*. (**B**) IC_50_ values of **1**–**3**; measuring compound concentration resulting in 50% inhibition of NO production in RAW 264.7 cells. (**C–E**) Cytostatic activity of **1**–**3**; evaluating compound effect on the proliferation rate of RAW 264.7 cells. Proliferation rate was measured using MTT tetrazolium dye, it was normalized to initial cells plated and compared to cells treated with carbowax 400 0.1% *v/v* + 0.01% ethanol. Statistical analysis was carried out using a Mann–Whitney unpaired t-test in Graphpad Prism 7.0 and graphs represent mean ± SEM (* indicates *p* < 0.05, ** indicates *p* < 0.01, *** indicates *p* < 0.001).

**Figure 2 marinedrugs-18-00527-f002:**
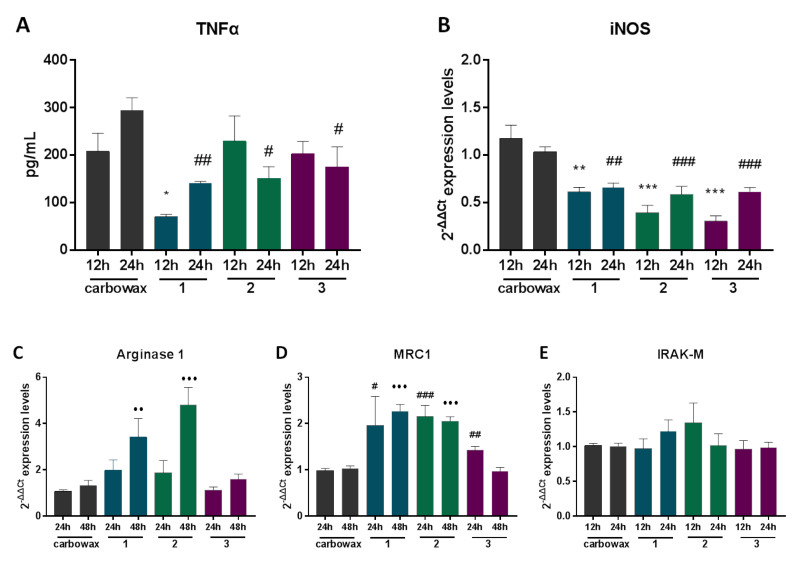
The effect of metabolites **1**–**3** on the expression of inflammatory markers in RAW 264.7 cells post 12, 24 and 48 h incubation with the indicated compound. (**A**) TNFα production was measured using ELISA in cell culture supernatants and (**B**) inducible nitric oxide synthase (iNOS) (**C**) Arginase 1 (**D**) MRC1 and (**E**) IRAK-M mRNA levels were identified using real time PCR. All treatments have been compared to carbowax 400 0.1% *v/v* + 0.01% ethanol treated cells in each timepoint indicated. Disulfide concentration used for **1** and **2** treatments was 15.62 μΜ and for **3** it was 31.25 μΜ. Statistical analysis was carried out using a Mann–Whitney unpaired t-test in Graphpad Prism 7.0 and graphs represent mean ± SEM and (* indicates *p* < 0.05, ** indicates *p* < 0.01, *** indicates *p* < 0.001 compared to carbowax 12 h, # indicates *p* < 0.05, ## indicates *p* < 0.01, ### indicates *p* < 0.001 compared to carbowax 24 h, ● indicates *p* < 0.05, ●● indicates *p* < 0.01, ●●● indicates *p* < 0.001 compared to carbowax 48 h).

**Figure 3 marinedrugs-18-00527-f003:**
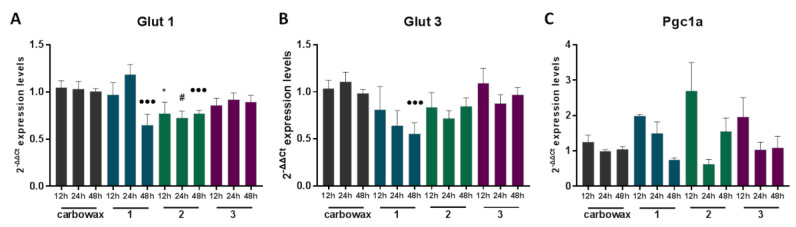
The effect of metabolites **1**–**3** on the expression of metabolic genes in RAW 264.7 macrophages following 12, 24 h and 48 h incubation with the indicated disulfide. (**A**) Glut 1, (**B**) Glut 3 and (**C**) Pgc1a expression was measured using real time PCR. All treatments have been compared to carbowax 400 0.1% *v/v* + 0.01% ethanol treated cells in each timepoint indicated. Disulfide concentration used for **1** and **2** treatments was 15.62 μΜ and for 3 it was 31.25 μΜ. Statistical analysis was carried out using a Mann–Whitney unpaired t-test in Graphpad Prism 7.0, and the graphs represent mean ± SEM (* indicates *p* < 0.05, ** indicates *p* < 0.01, *** indicates *p* < 0.001 compared to carbowax 12 h, # indicates *p* < 0.05, ## indicates *p* < 0.01, ### indicates *p* < 0.001 compared to carbowax 24 h, ● indicates *p* < 0.05, ●● indicates *p* < 0.01, ●●● indicates *p* < 0.001 compared to carbowax 48 h).

**Figure 4 marinedrugs-18-00527-f004:**
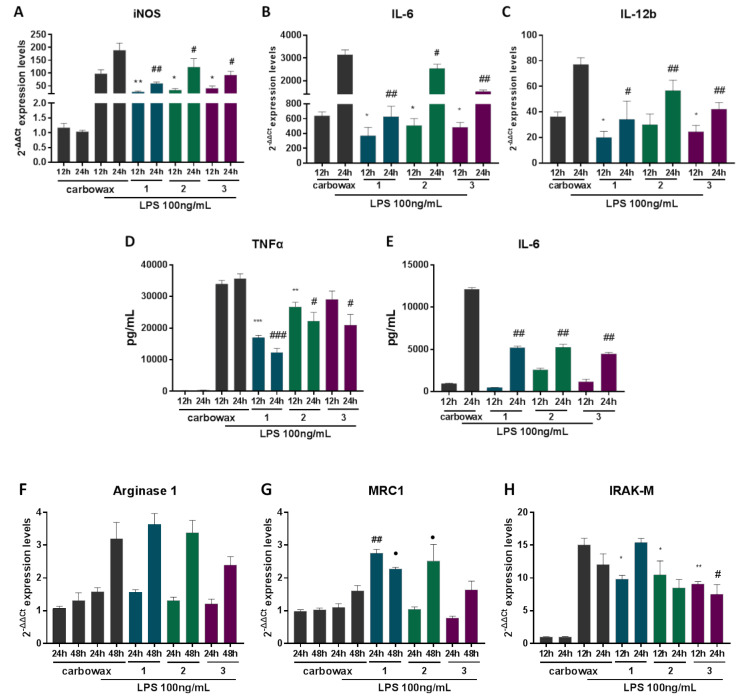
The effect of metabolites **1**–**3** on the expression of inflammatory genes in RAW 264.7 macrophages post LPS stimulation for 12, 24 and 48 h and simultaneous incubation with the respective disulfide. Pro-inflammatory markers (**A**) iNOS, (**B**) IL-6, (**C**) IL-12 mRNA expression levels were measured using real time PCR. (**D**) TNFα and (**E**) IL-6 production was determined using ELISA in cell culture supernatant. Markers of M2 macrophages (**F**) Arginase 1, (**G**) MRC1 and (**H**) IRAK-M expression levels were determined using real time PCR. All treatments have been compared to carbowax 400 0.1% *v/v* + 0.01% ethanol treated cells in each time point indicated. Disulfide concentration used for **1** and **2** treatments was 15.62 μΜ and for 3 it was 31.25 μΜ. Statistical analysis was carried out using a Mann–Whitney unpaired t-test in Graphpad Prism 7.0 and graphs represent mean ± SEM (* indicates *p* < 0.05, ** indicates *p* < 0.01, *** indicates *p* < 0.001 compared to carbowax 12 h, # indicates *p* < 0.05, ## indicates *p* < 0.01, ### indicates *p* < 0.001 compared to carbowax 24 h, ● indicates *p* < 0.05, ●● indicates *p* < 0.01, ●●● indicates *p* < 0.001 compared to carbowax 48 h).

**Figure 5 marinedrugs-18-00527-f005:**
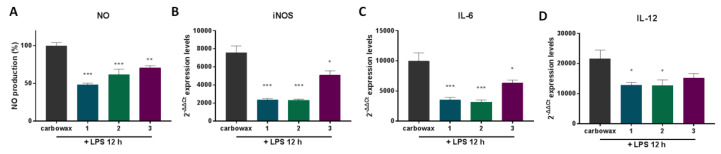
The effect of metabolites **1**–**3** on the expression of pro-inflammatory genes in primary thioglacolate-elicited macrophages post LPS stimulation for 12 h and simultaneous stimulation with the respective disulfide. (**A**) NO release in cell culture supernatant was measured using Griess reaction, (**B**) iNOS, (**C**) IL-6 and (**D**) IL-12 mRNA levels were quantified using real-time PCR. All treatments have been compared to carbowax 400 0.1% *v/v* + 0.01% ethanol treated cells in each time point indicated. Disulfide concentration used for **1** and **2** treatments was 15.62 μΜ and for **3** it was 31.25 μΜ. Statistical analysis was carried out using a Mann–Whitney unpaired t-test in Graphpad Prism 7.0 and graphs represent mean ± SEM (* indicates *p* < 0.05, ** indicates *p* < 0.01, *** indicates *p* < 0.001 compared to carbowax 12 h).

**Figure 6 marinedrugs-18-00527-f006:**
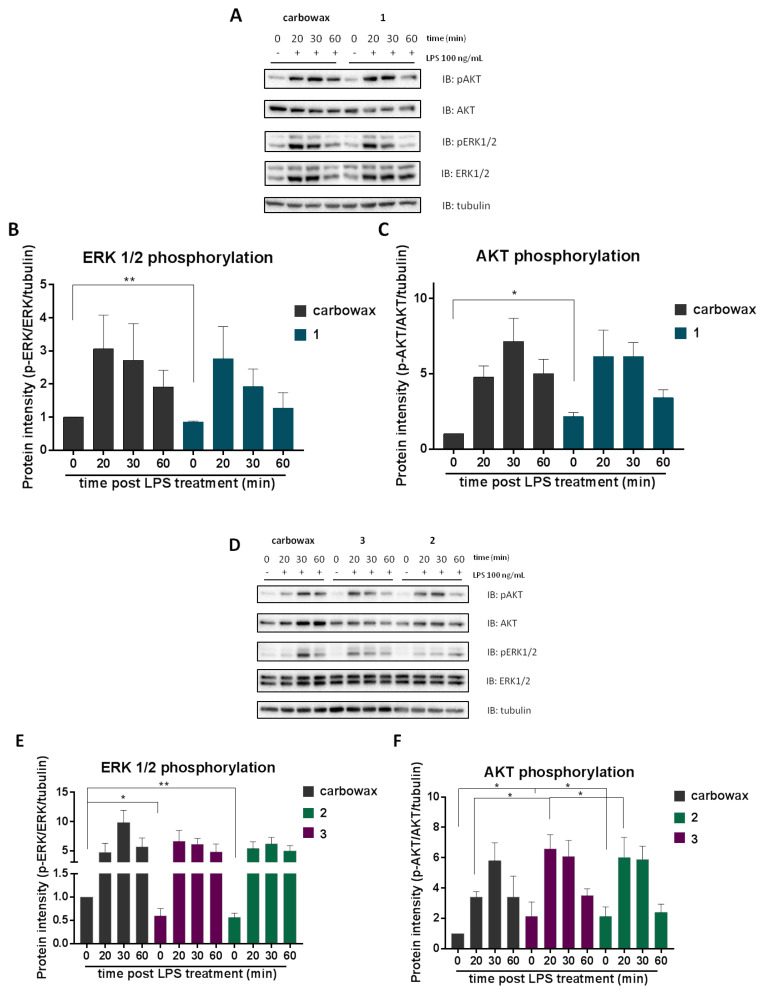
Monitoring the effect of metabolites **1**–**3** on AKT/ERK/MAPK signaling. RAW 264.7 macrophages were pre-incubated for 1 h with the respective disulfide and then activated for 20, 30 or 60 min with 100 ng/mL LPS. Cell lysates were electrophoresed in Western blot (**A**,**D**). Analysis of band intensity was quantified using Image Lab and compared to carbowax 400 0.1% *v/v* + 0.01% ethanol treated cells (**B**,**C**) and (**E**,**F**). Disulfide concentrations used for **1** and **2** treatments was 15.62 μΜ and for 3 it was 31.25 μΜ. Statistical analysis was carried out using a Mann–Whitney unpaired t-test in Graphpad Prism 7.0 and graphs represent mean ± SEM (* indicates *p* < 0.05, ** indicates *p* < 0.01, *** indicates *p* < 0.001 compared to carbowax 400).

**Table 1 marinedrugs-18-00527-t001:** IC_50_ values in μΜ for 50% inhibition of NO release and cytostatic activity of compounds **1**–**3**.

Compound	IC_50_ (μΜ)	No Cytostatic Activity (μΜ) (at 48 h)
**1**	2.6 ± 0.35	<62.5
**2**	3.8 ± 0.29	<45
**3**	14.2 ± 1.67	<125

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
