# Peer review of "Disulfides from the Brown Alga Dictyopteris membranacea Suppress M1 Macrophage Activation by Inducing AKT and Suppressing MAPK/ERK Signaling Pathways"

_marinedrugs, 2020, doi:10.3390/md18110527_

Round 1
Reviewer 1 Report
Please also see the attached file:
Figure 1B needs to be modified to a form that clearly shows the IC50 values in relation to the concentration data (preferably not using a logarithmic scale).
To be useful, Figures 1C-1E should also be adapted to a form where one axis shows concentrations and the other presents proliferation as a % of the proliferation of the control at any specific time. So you could replace these with 2 diagrams, each showing data for the 3 compounds tested: one for a 48 hour period and the other for 72 hr period, and you would be looking for the concentration of proliferated cells in the treated samples to drop to half that of the control group for IC50 values.
What I can tell from your current graphs (measuring distances) is that:
1 None of the 3 compounds at any of the concentrations graphed have achieved IC50 values after a 48 hr expt.
2. After 72 hours, compounds 1 at 62.5 mcg/L and compound 3 at 125 mcg/L are both essentially at the IC50 values, but 2 is still well short at the highest concentration you tested. So, 2 is less active than 1, not more active as your Table 1 suggests from whatever you did to the 48 hr data.

Author Response
Response to the comments of Reviewer 1
Comment: This manuscript details the biochemical metabolic effects caused by three sulphur-containing metabolites that have previously been reported as constituents with anti-inflammatory activity from the brown alga Dictyopteris membranacea and proposes the metabolite mode of action. It is mainly well written, and without typographical errors, but there are some calculated parameters where the results are not supported by the accompanying Figures (see below).
Response: We thank the reviewer for the positive view of our work and the constructive comments and suggestions.
1.Comment: Page 2 lines 55 and 59: I suggest "(pro-inflammatory)" and "(anti-inflammatory)" be inserted after Th1 and Th2 respectively, as these abbreviations are otherwise undefined.
Response: We thank the reviewer for the comment. “pro-inflammatory” has been inserted in line 55 and “anti-inflammatory” has been inserted in line 59.
2.Comment: lines 62-64: I suggest “Among them, the PI3K/AKT signaling cascade, which is involved in macrophage polarization, cell survival and proliferation, as well as cellular metabolism through direct interaction with mTORC2 plays a vital role [4-7]” to improve the English.
Response: We apologize the reviewer for the confusing language. Lines 62-64 “Among them, vital role holds the PI3K/AKT signaling cascade, which is involved in macrophage polarization, cell survival and proliferation, as well as cellular metabolism through direct interaction with mTORC2 [4-7]” has been replaced with “Among those, central role plays the PI3K/AKT signaling cascade. PI3K/AKT regulates macrophage polarization, cell survival, proliferation, as well as cellular metabolism [4-7].”
3.Comment:line 78: “…derivatised C11 hydrocarbons. (as the hydrocarbons themselves aren’t responsible for the odour).
Response: We thank the reviewer for the comment. Line 78 “the presence of” has been replaced with “derivatised”.
4.Comment: Page 3 Figure 1B: The colours used for graphs relating to compounds 2 and 3 here are essentially indistinguishable.
Response: We apologize to the reviewer for the confusing Figure 1B. We have modified colors and symbols to make the figure easier to follow.
5.Comment: line 111 and Table 1: There is a major problem with these IC50 values in relation to Figure 1B. For starters, the figure has a log scale on the x-axis, so almost half of the diagram contains no measured data. Moreover, assuming the concentrations used for IC50 (NO inhibition) here were the same as those for the cytotoxicity studies (Figs 1C-1E) the least concentrated sample was 3.9 μM (for 1). (Maybe this is an incorrect assumption, since Figs 1C-1E only used 5 concentrations and more data points are plotted in figure 1B. However, with a log scale that only has 0,1 and 100 as reference points, I cannot estimate the positions of the data point concentrations). All 3 lines in Figure 1B have at least one data point above 50% NO production, yet here you state an IC50 for 1 that is lower that than the lowest concentration of the solutions graphed (which clearly isn't the case in Figure 1B as the 50% value is clearly within the region graphed!). The same argument is true for 2 and 3, where the lowest concentrations presumably were 11.1 and 21.5 μM respectively, but both IC50 values have been stated here to be lower than those values!
Response: We apologize to the reviewer for the confusion. Figure 1B has been modified to make it more clear. As correctly noticed, the concentrations used for the cytostatic measurements in figures 1C-E are fewer than the ones used for the IC50 determination, since the purpose of the experiment was to ensure that the concentration of the compounds used in the analyses did not interfere with cell proliferation and that the anti-inflammatory action was not due to cytostatic effects. In addition compound 1 was found to be more active as the concentrations used in IC50 determination for compound 1 were: 1.95μΜ, 3.90μΜ, 7.81μΜ, 15.62μΜ, 31.25μΜ, 62.5μΜ. For compound 2 were: 2.23μΜ, 4.46μΜ, 6.69μΜ, 8.92μΜ, 13.38μΜ, 17.84μΜ, 22.33μΜ, 33.45μΜ, 44.60μΜ, 64.67μΜ. For compound 3 were: 4.31μΜ, 8.62μΜ, 12.93μΜ, 17.24μΜ, 25.86μΜ, 34.48μΜ, 43.1μΜ, 64.65μΜ, 86.20μΜ, 125μΜ. We have now plotted the graph to clearly illustrate the different concentrations used for each compound.
6.Comment: Page 4 Table 1: What is being used to determine whether these compounds were cytotoxic or not? Measurement of proliferation rates normally indicates cytostatic activity (inhibition of cell division) rather than cytotoxicity (where cells are being killed). The values given here appear different to what the Figures 1C-1E indicate: presumably you are comparing "proliferation rates" of treated samples to controls as indicated in the experimental details. For compounds 1 and 2, your figures 1C and 1D appear to have similar values for both compounds (at the highest concentration which is the closest to your calculated cytotoxicity) and controls after 48 hours, so 2 being significantly more active than 1 surprises me; 3 is obviously less active.
Response: We thank the reviewer for the comment and we apologize for using the wrong term. In Page 4 table 1 “Cytotoxicity” has been replaced with “Cytostatic Activity”. Line 102 (figure 2 legend) and line 114 : “Cytotoxic” has been replaced with “cytostatic” . Line 116 and 121 “Cytotoxic” has been replaced with “cytostatic potential”. Compound 1 has been stated to be slightly more active than compound 2 both concerning IC50 values and activation markers’ profile. In Figure 1C-D the highest concentration values used are listed below: For compound 1 after 48h the value of the proliferation rate in cells treated with 62.5 μΜ is 2.98; and for the control is 3.68. For compound 2 after 48h the value of the proliferation rate in cells treated with 64.7μΜ is 2.92 and for the control is 3.91.
7.Comment: Page 10 lines 308-310: Compound concentrations presented in this manuscript need to be significantly clarified in relation to Figure 1B and the IC50 issues previously mentioned. The statement here was presumably only true for the macrophage/gene suppression sections 2.2-2.4 and has nothing to do with the IC50 or cytotoxicity experiments
Response: We apologize for not stating the concentrations clearly. We have now included in the Materials and Methods the exact concentrations used for treatment of macrophages. Accordingly, page 10 lines 312-312 “Compounds’ concentrations used for treatment were approximately 3 times higher than the IC50 value of each compound” has been replaced with “Compounds’ concentrations used for macrophage treatments in figures 2-5 were 15.62 μΜ for compounds 1 and 2 and 31.25μΜ for compound 3.”
Reviewer 2 Report
Authors Daskalaki et al. present MS entitled “Disulfides from the brown alga Dictyopteris 2 membranacea suppress M1 macrophage activation by 3 inducing AKT and suppressing MAPK/ERK signaling 4 pathways” that they provide comprehensive analysis of modulatory effects of three selected disulfides isolated from Dictyopteris membranace on mouse macrophage cell line RAW 264.7. Data show that these tested disulfides a suppressed pro-inflammatory M1 macrophage phenotype (production of pro-inflammatory cytokines TNFα, IL-6 and IL-12, the expression of nitric oxide synthase) and enhanced markers of the M2 macrophage regulatory phenotype (activation of Arginase1 and MRC1). Based on analysis of AKT/MAPK/ERK signaling pathway modulation authors conclude that the effects were mediated by elevated AKT phosphorylation and MAPK/ERK signal transduction reduction. Authors in conclusion suggest that tested disulfides can be considered as potent candidates for the development of novel anti-inflammatory molecules with homeostatic properties.
The MS has limited novelty since these authors already shown that disulfides isolated from Dictyopteris membranacea suppress nitric oxide production from activated macrophages and have anti-inflammatory actions (Dimou, M.; Ioannou, E.; Daskalaki, M.G.; Tziveleka, L.A.; Kampranis, S.C.; Roussis, V. Disulfides with anti-inflammatory activity from the brown alga dictyopteris membranacea. J Nat. Prod. 2016, 79, 584-589). However, the present study widens the primary observation by comprehensive analysis of regulatory effects on differentiation of macrophages into M1 and M2 subpopulations. However, the major limitation is the application of only stable cell line RAW 264.7. To confirm the results with primary human cells would significantly improve the informative value of the results.
Comments:
• In generally, untreated naïve RAW 264,7 macrophages should not express significant levels of iNOS or produce NO. Only after stimulation by e.g. LPS. Thus, it should be better clarify if the cells were activated by LPS and consequently treated by tested compounds. E.g. row 130 states that macrophages were LPS stimulated. Next sentence state that the macrophages were naïve. This should be also highlighted in the figure legends to make them self-explanatory.
• The part of the text (row 158 -163) does not fit to the context of rest of the text. mTOR expression was not determined in the presented work. Or do authors suggest that mTOR is responsible for GLUT expression in naïve RAW 264.7 cells? This part should be more focused on regulation of expression of glucose transporters in naïve macrophages.
• Since tested compounds are suggested to potentiate expression of GLUT transporters, and at the same time, authors show only downregulation of selected signaling pathways, it is necessary to suggest what is a driving signaling machinery for the GLUTs upregulation.
• The statement (page 6, row 208) “We showed that although M1 activation was impaired in compound-treated cells, M2 markers were induced at later time points.” should be rewritten.
• In Figure 1 C. The upper double star significance is unclear
• Row 130 iNOS is Inducible nitric oxide synthase
• It would be better to name Arginase 1, MRC1 and IRAK-M as markers of M2 macrophages than Anti-inflammatory markers e.g. in figure legend 4.
• Page 7 row 234-237 - neither HIF nor STAT are typical downstream signaling partners of MAPK kinases. These statements could be omitted.
• Final Conclusions (rows 274- 277) - the presented data do not support the conlusion that “This action is mediated by suppression of glucose transporters …”. Much more deeper analysis of cell metabolism would have to be performed to support this conclusions. Thus the conclusions should be rewritten.
Author Response
Response to the comments of Reviewer 2
Comment: Authors Daskalaki et al. present MS entitled “Disulfides from the brown alga Dictyopteris 2 membranacea suppress M1 macrophage activation by 3 inducing AKT and suppressing MAPK/ERK signaling 4 pathways” that they provide comprehensive analysis of modulatory effects of three selected disulfides isolated from Dictyopteris membranace on mouse macrophage cell line RAW 264.7. Data show that these tested disulfides a suppressed pro-inflammatory M1 macrophage phenotype (production of pro-inflammatory cytokines TNFα, IL-6 and IL-12, the expression of nitric oxide synthase) and enhanced markers of the M2 macrophage regulatory phenotype (activation of Arginase1 and MRC1). Based on analysis of AKT/MAPK/ERK signaling pathway modulation authors conclude that the effects were mediated by elevated AKT phosphorylation and MAPK/ERK signal transduction reduction. Authors in conclusion suggest that tested disulfides can be considered as potent candidates for the development of novel anti-inflammatory molecules with homeostatic properties.
The MS has limited novelty since these authors already shown that disulfides isolated from Dictyopteris membranacea suppress nitric oxide production from activated macrophages and have anti-inflammatory actions (Dimou, M.; Ioannou, E.; Daskalaki, M.G.; Tziveleka, L.A.; Kampranis, S.C.; Roussis, V. Disulfides with anti-inflammatory activity from the brown alga dictyopteris membranacea. J Nat. Prod. 2016, 79, 584-589). However, the present study widens the primary observation by comprehensive analysis of regulatory effects on differentiation of macrophages into M1 and M2 subpopulations. However, the major limitation is the application of only stable cell line RAW 264.7. To confirm the results with primary human cells would significantly improve the informative value of the results.
Response: We thank the reviewer for the positive view of our work and the constructive comments and suggestions. According to the suggestion of the reviewer, we have performed additional experiments using primary mouse macrophages to confirm the key findings in Raw264.7 macrophages. Use of human macrophages was not possible since it required licences that we did not have available. The results are now included in figure 5 of the revised manuscript.
Comments:
1.Comment: In generally, untreated naïve RAW 264,7 macrophages should not express significant levels of iNOS or produce NO. Only after stimulation by e.g. LPS. Thus, it should be better clarify if the cells were activated by LPS and consequently treated by tested compounds. E.g. row 130 states that macrophages were LPS stimulated. Next sentence state that the macrophages were naïve. This should be also highlighted in the figure legends to make them self-explanatory.
Response: We apologize to the reviewer for the confusion. In line 133 “in LPS activated macrophages” was replaced with “non-activated macrophages”. As it is stated in Figure 2 legend, macrophages were non stimulated and were only compound treated. RAW 264.7 macrophage cell line expresses low but measurable levels of iNOS mRNA.
2.Comment: The part of the text (row 158 -163) does not fit to the context of rest of the text. mTOR expression was not determined in the presented work. Or do authors suggest that mTOR is responsible for GLUT expression in naïve RAW 264.7 cells? This part should be more focused on regulation of expression of glucose transporters in naïve macrophages.
Response: We thank the reviewer for the comment. In lines 160-161 “The mammalian target of rapamycin (mTOR) exhibits a plethora of cellular functions, including controlling metabolic pathways and shaping the magnitude of the immune response [16]” was deleted according to the Reviewer’s suggestion.
3.Comment: Since tested compounds are suggested to potentiate expression of GLUT transporters, and at the same time, authors show only downregulation of selected signaling pathways, it is necessary to suggest what is a driving signaling machinery for the GLUTs upregulation.
Response: We apologize for not stating the action of the compounds clearly. Our data show that GLUTs are down regulated in response to compound treatments and do not potentiate them. We have rephrased the sentence explaining the role of GLUTs in macrophages (line 193).
4.Comment:The statement (page 6, row 208) “We showed that although M1 activation was impaired in compound-treated cells, M2 markers were induced at later time points.” should be rewritten.
Response: We apologize to the reviewer for the confusion. In lines 211-213 “We showed that although M1 activation was impaired in compound-treated cells, M2 markers were induced at later time points” was replaced with “We showed that although M1 activation was impaired in compound-treated cells, M2 markers, which were induced upon LPS stimulation at 24 and 48 hours as a negative feedback mechanism, were not affected by the compounds.”
5.Comment: In Figure 1 C. The upper double star significance is unclear
Response: We thank the reviewer for the comment. Figure 1C was modified accordingly.
6.Comment: Row 130 iNOS is Inducible nitric oxide synthase
Response: We thank the reviewer for the comment. In Row 131 “Inducer of nitric oxide” was replaced with “Inducible nitric oxide synthase”
7.Comment: It would be better to name Arginase 1, MRC1 and IRAK-M as markers of M2 macrophages than Anti-inflammatory markers e.g. in figure legend 4.
Response: We thank the reviewer for the comment. In line 224 (Figure 4 legend) “Anti-inflammatory markers” was deleted and “Markers of M2 macrophages was inserted”.
8.Comment: Page 7 row 234-237 - neither HIF nor STAT are typical downstream signaling partners of MAPK kinases. These statements could be omitted.
Response: We thank the reviewer for the comment. Lines 236-239 “ERK1/2 is a member of the MAPK signaling pathway, which is phosphorylated in response to various signals and translocates from the cytosol to the nucleus, activating several transcription factors, including Hypoxia-inducible-factor-1 (HIF-1) and signal transducer and activator of transcription-3 (STAT3)” have been deleted.
9.Comment: Final Conclusions (rows 274- 277) - the presented data do not support the conclusion that “This action is mediated by suppression of glucose transporters …”. Much more deeper analysis of cell metabolism would have to be performed to support this conclusions. Thus the conclusions should be rewritten.
Response: We agree with the reviewer that the data presented only support a suppressive action of the compounds on Glut1 and Glut3 and do not prove that their anti-inflammatory effect is mediated through this action. We have, therefore, rephrased the conclusion as follows: “This action may be partly due to suppression of the expression of glucose transporters, thus potentially reducing the glucose availability and glycolytic capacity of the cells. GLUT1 and GLUT3 expression are induced by different signals such as IGF1 or cytokines, leading to NFkB activation PMID: 31447391. It is therefore likely that compounds directly or indirectly affect GLUT1 and GLUT3 transcription. In addition, the compounds also suppressed the pro-inflammatory signaling cascade of MAPK, while induced the anti-inflammatory signaling cascade of AKT, providing an additional potential mechanism for their anti-inflammatory action. Further studies are required to confirm these mechanisms.”
Reviewer 3 Report
This manuscript written by Daskalaki et al. describes that three disulfides isolated from the brown alga Dictyopteris membranacea show the anti-inflammatory activity by suppressing the expression of proinflammatory cytokine genes and inducing the phosphorylation of Akt. The data look interesting. This reviewer thinks that this paper would attract the attention of the readers of Marine Drugs. There are several concerns about this paper as shown below.
Major comments
In Figure 5, statistical significance of the difference in the phosphorylation state of ERK and AKT is mainly observed at zero minute after LPS treatment; however, the phosphorylation level of ERK and AKT at this time point is quite low because the RAW 264.7 cells have not been stimulated with LPS. Actually, the bands of phosphorylated AKT and ERK look very faint in the picture. Thus, this reviewer does not understand how important these significant differences are and what these mean. The authors need to address these questions in the discussion section.
Did the three disulfides coexist in the culture medium with LPS in the pretreatment experiments? In other words, did the authors wash each well and then add disulfides or LPS after the pretreatment? If so, is it possible to think that the three compounds do not affect RAW 264.7 cells but interact directly with LPS in the medium to cancel the effect of LPS on the cells?
Why was the TNF concentration lower at 24 h than at 12 h in Figure 2A and 4D? In general, a cytokine secreted from cells accumulates in the medium, and its concentration in the medium increases depending on the incubation period. The authors need to discuss the possible reason.
How is the gene expression of GLUT-1 regulated in macrophages? The authors need to discuss the possible mechanism of action of the compounds 1 and 2 on the downregulated gene expression of GLUT-1.
The Mann–Whitney U test can be used to compare two groups; however, the authors compared more than two groups in all experiments conducted in this study. Thus, the authors have to use a multiple comparison test, such as Tukey's test.
Minor comments
Line 299: Show the purity of these three disulfides.
Line 323: Show the seeded cell density.
Author Response
Response to the comments of Reviewer 3
Comment: This manuscript written by Daskalaki et al. describes that three disulfides isolated from the brown alga Dictyopteris membranacea show the anti-inflammatory activity by suppressing the expression of proinflammatory cytokine genes and inducing the phosphorylation of Akt. The data look interesting. This reviewer thinks that this paper would attract the attention of the readers of Marine Drugs. There are several concerns about this paper as shown below.
Response: We thank the reviewer for the positive view of our work and the constructive comments and suggestions.
Major comments
1.Comment: In Figure 5, statistical significance of the difference in the phosphorylation state of ERK and AKT is mainly observed at zero minute after LPS treatment; however, the phosphorylation level of ERK and AKT at this time point is quite low because the RAW 264.7 cells have not been stimulated with LPS. Actually, the bands of phosphorylated AKT and ERK look very faint in the picture. Thus, this reviewer does not understand how important these significant differences are and what these mean. The authors need to address these questions in the discussion section.
Response: We thank the reviewer for the comment. Indeed, the only significant differences in phosphorylation levels concern basal phosphorylation. Nevertheless, we hypothesize that the differences observed in naïve state may be responsible for the significant differences observed in non-activated macrophages. Although RAW 264.7 macrophages express low levels of pro-inflammatory markers in non-activated state we hypothesize that they are primed to enter a hyporesponsive state due to treatment with the compounds.
2.Comment: Did the three disulfides coexist in the culture medium with LPS in the pretreatment experiments? In other words, did the authors wash each well and then add disulfides or LPS after the pretreatment? If so, is it possible to think that the three compounds do not affect RAW 264.7 cells but interact directly with LPS in the medium to cancel the effect of LPS on the cells?
Response: We thank the reviewer for the insightful comment. The wells were not washed after the pretreatment, therefore the compounds co-existed with the LPS. It is possible that compounds could directly interact with LPS acting as scavenger and neutralizing its effect as it is has been shown in disulfides acting as defensins (Ref 20). Nevertheless, since we have shown that tested disulfides exhibit significant anti-inflammatory actions in non-activated macrophages, it is less likely that they would also exhibit direct binding to LPS. This possibility is discussed in lines 233-236.
3.Comment: Why was the TNF concentration lower at 24 h than at 12 h in Figure 2A and 4D? In general, a cytokine secreted from cells accumulates in the medium, and its concentration in the medium increases depending on the incubation period. The authors need to discuss the possible reason.
Response: We thank the reviewer for the comment. In figure 2A we measure the basal levels of TNFa production in culture medium which is as expected (non-activated macrophages) low (50-300 pg/mL). Compounds 2 and 3 do not exhibit significant effect on TNF levels in 12 and 24 hours. In figure 4D, we measured the protein levels of TNFa in LPS-activated cells. TNF is lower at the 24h timepoint compared to the 12h timepoint, since TNFa starts being produced early and accumulates in the supernatant of the culture through time but some of it may become degraded thorough time. Therefore the accumulated TNFa may be lower at 24 hours than 12 hours.
4.Comment:How is the gene expression of GLUT-1 regulated in macrophages? The authors need to discuss the possible mechanism of action of the compounds 1 and 2 on the downregulated gene expression of GLUT-1.
Response: We thank the reviewer for the comment. Please check response to comment 9 of reviewer 2.
5.Comment:The Mann–Whitney U test can be used to compare two groups; however, the authors compared more than two groups in all experiments conducted in this study. Thus, the authors have to use a multiple comparison test, such as Tukey's test.
Response: We thank the reviewer for the comment. Although there are different groups tested in each experiment, each timepoint and treatment is compared with the respective control (carbowax) at the indicated timepoint, therefore we chose Mann–Whitney U test. Nevertheless, we repeated all statistical analyses using Tukey’s test resulting in similar results. The analysis is now commented in line 461.
Minor comments
6.Comment:Line 299: Show the purity of these three disulfides.
Response: He have now provided data indicating the purity of the compounds (supplemental figures 1-3).
7.Comment:Line 323: Show the seeded cell density.
Response: We thank the reviewer for the comment. The number of cells seeded, “5 x 104”, was inserted in line 397.
Round 2
Reviewer 1 Report
Please see the attached file.

Author Response
Comment 1: Page 2 lines 63: I previously suggested relocating the phrase about the central role of the signaling cascade to improve the English, but that has not been done: “Among those, central role plays the PI3K/AKT signaling cascade” is incorrect English and should be modified to: “Among those, the PI3K/AKT signaling cascade plays a central role”
Response: We thank the reviewer for the suggestion and we apologize again for the incorrect English. In Page 2 line 63 the phrase “Among those, central role plays the PI3K/AKT signaling cascade” has been replaced with Among those, the PI3K/AKT signaling cascade plays a central role in macrophage polarization”
Comment 2: Page 3 line 96 Figure 1B: This Figure is now much improved, but the number in IC50 should be a subscript and the scale should include 0.2 at the x-origin.
Response: We thank the reviewer for the comment. In page 3 Figure 1B, the IC50 number has been corrected and 0.2 has been indicated on the x-axis.
Comment 3: Page 4 Table 1: I still have a problem with how the values for the “cytostatic activity” in this Table were obtained from Figures 1C-E. While I understand that these graphs have been “normalized” to the same number of cells at the start (arbitrarily shown as “1” on the graphs), the author’s response to my previous comments does not truly represent the growth (or proliferation) rate of the cells. You say: “For compound 1 after 48h the value of the proliferation rate in cells treated with 62.5 μΜ is 2.98; and for the control is 3.68. For compound 2 after 48h the value of the proliferation rate in cells treated with 64.7μΜ is 2.92 and for the control is 3.91.” Although the data point on the graph after 48hr for the control and for 1 are respectively 3.68 and 2.98, the increases in the number of cells (from when the experiment was started, i.e. the proliferation) are only 2.68 and 1.98 times the number of cells you started with (since you started from 1), so a 26% decrease for the treatment. Similarly, for 2 after 48 hr, the increases in the number of cells (from when the experiment was started, i.e. the proliferation) are 2.91 and 1.68 times the number of cells you started with (since you started from 1), so a 43% decrease. There was only a 3.5% difference in the concentrations, so since neither 1 or 2 has reached 50% growth inhibition in these maximum concentration tested assays, how can the cytostatic potential be lower than the maximum concentration used for 2 (even though 2 is slightly more cytostatic than 1)?
Response: We apologize for not clarifying the purpose of the experiment and the findings, which confused the reviewer. The purpose for determining cytostatic activity of the compounds is to determine the concentrations at which no cytostatic activity was present. We did not want to evaluate the cytostatic potential of the compounds and we did not aim to compare them. Therefore, we used concentrations that at a 48 hour treatment did not exhibit any cytostatic action on the cells. We have modified the manuscript to clarify the purpose of the experiment and avoided any comparison of the cytostatic potential of the compounds. In the table we now mention clearly that the concentrations shown are those that no cytostatic activity is evident, to avoid the reader comparing the cytostatic potential between compounds.
Comment 4: Page 7 line 224: “respected” should be “respective”. (The meanings of these two words differ).
Response: We apologize to the reviewer for the mistake. In line 225, line 236 and line 245 “respected” has been replaced with “respective”.
Reviewer 2 Report
1- I still see the down regulation of the GLUT transporters as only a minor effect that could not explain the inhibitory effects of disulfides. For example, the compound 3 does not decrease GLUT transporters at all. Thus could not explain the inhibitory effect on M1 activation of macrophages.
I would suggest that conclusions both in the abstract and in the mien text should be modified accordingly.
2- Newly added sentence “We showed that although M1 activation was impaired in compound-treated cells, M2 markers, which were induced upon LPS stimulation at 24 and 48 hours as a negative feedback mechanism, were not affected by the compounds.” does not reflect the results. The expression was altered since .. .. All three disulfides did not affect expression of the M2 marker Arginase 1 (Figure 4F), whereas MRC1 was significantly up-regulated in activated macrophages treated with disulfides 1 and 2 (Figure 4G).
3 It would be helpful to mention what the tested concentrations of compounds were in the figure legends.
Author Response
Comment 1: I still see the down regulation of the GLUT transporters as only a minor effect that could not explain the inhibitory effects of disulfides. For example, the compound 3 does not decrease GLUT transporters at all. Thus could not explain the inhibitory effect on M1 activation of macrophages. I would suggest that conclusions both in the abstract and in the mien text should be modified accordingly.
Response: We thank the reviewer for the insightful comment. We have now changed all statements that may suggest causality between metabolic changes and anti-inflammatory activity of disulfides. Therefore, lines 34-35 commenting GLUT results in the abstract have been modified. In accordance, lines 160-184 have been modified and the comment in the conclusions has been removed.
Comment 2: Newly added sentence “We showed that although M1 activation was impaired in compound-treated cells, M2 markers, which were induced upon LPS stimulation at 24 and 48 hours as a negative feedback mechanism, were not affected by the compounds.” does not reflect the results. The expression was altered since .. .. All three disulfides did not affect expression of the M2 marker Arginase 1 (Figure 4F), whereas MRC1 was significantly up-regulated in activated macrophages treated with disulfides 1 and 2 (Figure 4G).
Response: We thank the reviewer for the suggestion. In lines 224-226 the sentence: “We showed that although M1 activation was impaired in compound-treated cells, M2 markers, which were induced upon LPS stimulation at 24 and 48 hours as a negative feedback mechanism, were not affected by the compounds” has been replaced with “We showed that although M1 activation was impaired in compound-treated cells, M2 markers were found to be induced upon LPS stimulation at 24 and 48 hours as a negative feedback mechanism.”
Comment 3: It would be helpful to mention what the tested concentrations of compounds were in the figure legends.
Response: We thank the reviewer for the comment. “Disulfide concentration used for 1 and 2 treatments was 15.62 μΜ and for 3 it was 31.25 μΜ” was added in each figure legends.